# Knowledge, attitude and practice of healthcare providers on mistreatment of women during labour and childbirth: A cross-sectional study in Tehran, Iran, 2021

**Marjan Mirzania[1], Elham Shakibazadeh[2]\*, Meghan A. Bohren[3], Farah Babaey[4], Sedigheh Hantoushzadeh[5], Abdoljavad Khajavi[6], Abbas Rahimi Foroushani[7]**

1 Department of Social Medicine, School of Medicine, Lorestan University of Medical Sciences, Khorramabad, Iran, 2 Department of Health Education and Promotion, School of Public Health, Tehran University of Medical Sciences, Tehran, Iran, 3 Gender and Women's Health Unit, Nossal Institute for Global Health, School of Population and Global Health, University of Melbourne, Carlton, Victoria, Australia, 4 Department of Midwifery, Ministry of Health and Medical Education, Tehran, Iran, 5 Department of Obstetrics and Gynecology, School of Medicine, Vali-E-Asr Reproductive Health research Center, Family Health Research Institute, Tehran University of Medical Sciences, Tehran, Iran, 6 Department of Social Medicine, School of Medicine, Gonabad University of Medical Sciences, Gonabad, Iran, 7 Department of Epidemiology and Biostatistics, School of Public Health, Tehran University of Medical Sciences, Tehran, Iran

\* shakibazadeh@tums.ac.ir

**Data Availability Statement:** All relevant data are within the paper.

## Abstract

### Background

Mistreatment of women during childbirth is a global health challenge. Maternity healthcare providers play a key role in influencing women's birth experience. This study aimed to assess the knowledge, attitudes, and practices of healthcare providers regarding mistreatment of women during labour and childbirth in public hospitals in Tehran, Iran.

### Methods

This cross-sectional study was part of an implementation research project that was conducted from October to December 2021 at five public teaching hospitals in Tehran. All eligible maternity healthcare providers (obstetricians and midwives) and students were invited to participate in this study. Data were collected using a questionnaire consisting of four sections: socio-demographic characteristics (11 items), knowledge (11 items), attitudes (13 items), and practices (14 items) about mistreatment. Knowledge, attitude, and practice scores were determined using Bloom's cut-off points. Logistic regression analyses were used to identify the socio-demographic characteristics associated with knowledge and attitudes. A p-value of <0.05 was considered statistically significant.

### Results

Of the 270 participants, 255 (94.5%) participated in the study. Majority of the participants (82.7%) had poor knowledge regarding mistreatment of women during labour and childbirth. Poor knowledge was more apparent in the categories of physical abuse, verbal abuse, poor

**Funding:** This work was supported by the Tehran University of Medical Sciences (TUMS) and Health Information Management Research Center, TUMS (grant numbers 1401-3-208-62407). The roles of the funders are to monitor the corresponding study planning and progression.

**Competing interests:** The authors have declared that no competing interests exist.

rapport between women and providers, and failure to meet professional standards of care. Most participants (69.4%) had poor attitudes towards mistreatment; they were alright with physical abuse, verbal abuse, and discrimination. Only 3.1% of the participants reported moderate mistreatment practices towards birthing women. Verbal and physical abuse were the most prevalent categories used by the participants. The number of night shifts was associated with attitudes regarding mistreatment (AOR = 0.45, 95% CI = 0.22–0.89, p = 0.02).

## Conclusion

The knowledge and attitude of our participants regarding maternity mistreatment were poor. A small percentage of the participants reported mistreatment practices. The findings of our study have important implications for program planners and decision-makers in developing effective interventions to reduce mistreatment of women during labour and childbirth in Iran.

## Introduction

Pregnancy and childbirth are everlasting experiences of women and their families worldwide. The concept of "safe motherhood", which has historically been limited to physical safety, has more recently expanded beyond the prevention of morbidity or mortality to encompass respect for women's fundamental human rights, including respect for their autonomy, dignity, feelings, choices, and preferences [1]. However, evidence from around the world presents a different and disturbing picture [1]. Mistreatment or disrespect and abuse (D&A) of women seeking maternity care is a global challenge that affects women's decisions on mode of birth, lactation, childbirth experience, childbirth progress, mother-child bonding, and mental health status, and violates their human rights [2, 3]. The World Health Organization (WHO) issued a statement calling for immediate attention to this challenge, stating that "every woman has the right to the highest attainable standard of health, including the right to dignified and respectful care throughout pregnancy, childbirth, postpartum period, as well as the right to be free from violence and discrimination" [4].

Mistreatment of women during childbirth is not a new issue. In the 1970s, following reports by women's health and rights advocates of mistreatment of women during labour and childbirth [5], the Humanization of Childbirth movement in Brazil was formed with an emphasis on promoting respectful maternity care (RMC) [6]. However, this movement lost its spotlight during the late 1990s [7]. In 2010, Bowser and Hill conducted a landscape analysis of the factors influencing the provision of RMC and seven categories of D&A during childbirth, including physical abuse, non-consented care, non-confidential care, non-dignified care, discrimination, abandonment or denial of care, and detention in facilities [8]. In 2015, Bohren et al. conducted a systematic review and proposed a modified typology of mistreatment, including physical abuse, sexual abuse, verbal abuse, stigma and discrimination, failure to meet professional standards of care, poor rapport between women and providers, and conditions and limitations of the health system [9].

According to the WHO, RMC refers to "the care provided to all women in a manner that maintains their dignity, privacy and confidentiality, ensures freedom from harm and mistreatment, and enables informed choice and continuous support during labour and childbirth" [10]. According to a qualitative evidence synthesis conducted by Shakibazadeh et al. in 2017, RMC comprises 12 domains, ranging from being free from harm and mistreatment to continuity of care [11]. However, despite the recognition of RMC as one of the basic rights of every

childbearing woman and an important component of quality of care, growing evidence shows that mistreatment during childbirth has become a common experience for women worldwide [12–15]. In a study of 38 hospitals and health centers in Ethiopia, 74% of women reported experiencing some form of mistreatment [16]. A systematic review of five studies in Ethiopia, Kenya, Nigeria, and Tanzania found that the prevalence of D&A ranged from 15 to 98% [17]. A WHO-led study (2019) in Ghana, Guinea, Nigeria and Myanmar found that 35.4% of women experienced physical or verbal abuse, or stigma or discrimination during childbirth [18].

Despite a significant reduction in the maternal mortality rate in Iran from 48 deaths per 100 000 live births in 2000 to 16 deaths in 2017 and the achievement of the fifth Millennium Development Goal (MDG 5) [19], the quality of maternity care remains suboptimal [20, 21]. Although research on mistreatment or respectful care is limited in Iran, there is evidence that mistreatment by women during labour and childbirth in hospitals is common, which may contribute to the low quality of maternity services. For example, in a 2019 study, 75.7% of women reported experiencing one or more types of mistreatment [2]. In another study, all women (n = 357) experienced at least one disrespectful behavior [22]. Moreover, the study of Shirzad et al. (2019) showed that mistreatment during maternity care in hospitals was the main reason for choosing cesarean section (CS) by Iranian women in a context where more than 50% of births take place by CS [23].

Consideration of the attitudes and behaviors of maternity healthcare providers (MHCPs) has been recognized as an effort to improve the quality of maternity care [24] because they play an important role in shaping the experiences (negative or positive) of pregnant and/or birthing women [25]. Based on the available evidence, poor attitudes and behaviors of MHCPs have significant impacts on care and the physical, psychological, and emotional health of women and their infants [24]. They are also important predictors of mistreatment and are key barriers to RMC [26, 27]. Gaps in the knowledge, attitudes, and behaviors of MHCPs regarding RMC have been documented in some studies [26–30], and the results of a systematic review showed that negative attitudes and behaviors of MHCPs undermine health care seeking and affect patient well-being and satisfaction with care [31]. The main step in designing interventions to prevent mistreatment is to assess the knowledge, attitudes, and practices (KAP) of MHCPs, with the goal of improving negative behaviors and reinforcing positive behaviors. To our knowledge, there is no comprehensive study in this field in Iran; therefore, this study sought to assess the KAP of healthcare providers regarding mistreatment of women during labour and childbirth in Tehran public hospitals.

## Methods

### Study design

This cross-sectional study is the first phase (needs assessment) of an implementation research project involving the development and implementation of a context-specific intervention to reduce disrespectful maternity care and evaluation of strategies to improve implementation. This phase was conducted in five public teaching hospitals in Tehran, Iran, between October and December 2021.

### Setting and participants

This study was conducted in Tehran, the capital city of Iran, with an area of more than 615 km$^2$ and a population of approximately 9 million [32]. Tehran has 18 public hospitals, 50 private hospitals, and 28 other hospitals (hospitals that are a subset of a specific organization, such as charity, affiliated with Armed Forces, affiliated with the Social Security Organization,

and affiliated with Islamic Azad University). The total number of deliveries in Iranian hospitals (2021) was 1163169, of which 54.3% were in public hospitals [33]. Potential participants in this study were MHCPs (obstetricians, midwives, obstetric residents, midwifery, and medical interns) who were recruited from five selected public hospitals in Tehran from October to December 2021. The characteristics and names of these hospitals are listed in the S1 Appendix.

MHCPs with the following characteristics were invited to the study: a) having at least one year of experience as obstetricians and midwives; b) passing at least one semester (six months) in the maternity ward for obstetric residents and midwifery students; and c) passing half of the internship (four weeks) in the maternity ward for medical interns' students.

### Sample size and sampling

The sample size was calculated using the formula (n $= \frac{(Z_{1-\frac{\alpha}{2}})^2 \times S^2}{d^2}$). Considering that the mean score of knowledge, attitudes, and practices of participants was calculated based on 100 (the calculation method is explained in the Methods section), the standard deviation (SD) of the score was 16.7, which was obtained by dividing the range of each score (from 0 to 100) by six SDs [34]. A sample size of 270 was determined, with a 95% confidence level and maximum error of two scores. The sampling frame of this study included all public hospitals with high delivery rates that provided a wide range of obstetrics services to women. From each of the northern, southern, eastern, western, and central districts of Tehran, a list of these hospitals was prepared (based on the Ministry of Health and Medical Education data [33]). A public hospital was randomly selected from each district (five hospitals in total). All eligible MHCPs working in the five hospitals were invited to participate in the study.

### Data collection

After obtaining ethical approval from the Tehran University of Medical Sciences (TUMS), the first author (MM) held meetings with officials (head of the hospitals and delivery units) in the hospitals to describe the objectives of the study and obtain approval. She received training on how to conduct interviews from the second author (ESh with experience in quantitative and qualitative research), obtained written informed consent from the participants, and collected the data through face-to-face interviews. The participants were assured that their information would be kept confidential and that they had the right to withdraw from the study at any time. Fifteen participants refused to participate in the study because of their high workloads and lack of sufficient time.

### Study measures

**Stage 1: Development of the questionnaire.** The data collection tool was a self-report written questionnaire in Persian, which was prepared using the typology of mistreatment of women during childbirth presented by Bohren et al. [9] (Table 1) and a literature review [35–40]. The questionnaire consisted of four sections. The first section included socio-demographic characteristics, including age, marital status, profession, field of study, monthly income, work experience, and number of night shifts. This section also included a question to identify the number of participants who had been trained in RMC: *"Have you ever been trained on RMC? If yes, what kind of training was it?"*. The second section consisted of 11 items that assessed the participants' knowledge of women's mistreatment during labour and childbirth. The third section consisted of 13 items that addressed participants' attitudes. The fourth

**Table 1. Typology of mistreatment of women during childbirth.**

| Categories of mistreatment | Examples |
|---|---|
| Physical abuse | Women beaten, slapped, kicked, or pinched during delivery; Women physically restrained to the bed or gagged during delivery |
| Sexual abuse | Sexual abuse or rape |
| Verbal abuse | Harsh or rude language; Judgmental or accusatory comments; Threats of withholding treatment or poor outcomes; Blaming for poor outcomes |
| Stigma and discrimination | Discrimination based on ethnicity/race/religion; Discrimination based on age; Discrimination based on socioeconomic status; Discrimination based on HIV status |
| Failure to meet professional standards of care | Lack of informed consent process; Breaches of confidentiality; Painful vaginal exams; Refusal to provide pain relief; Performance of unconsented surgical operations; Neglect, abandonment, or long delays; Skilled attendant absent at time of delivery |
| Poor rapport between women and providers | Poor communication; Dismissal of women's concerns; Language and interpretation issues; Poor staff attitudes; Lack of supportive care from health workers; Denial or lack of birth companions; Women treated as passive participants during childbirth; Denial of food, fluids and mobility; Lack of respect for women's preferred birth positions; Denial of safe traditional practices; Objectification of women; Detainment in facilities |
| Health systems conditions and constraints | Physical condition of facilities; Staffing constraints; Staffing shortages; Supply constraints; Lack of privacy; Lack of redress, Facility culture Bribery and extortion; Unclear fee structures; Unreasonable requests of women by health workers |

Resource: Bohren et al. (2015) The mistreatment of women during childbirth in health facilities globally: a mixed-methods systematic review [9].

section consists of 14 items that assess their practices. The KAP items included the categories of physical abuse, verbal abuse, stigma and discrimination, failure to meet professional standards of care, poor rapport between women and providers, and health systems conditions and constraints (S2 Appendix).

**Stage 2: Pilot testing of the questionnaire.** The validity and reliability of the questionnaire were evaluated at this stage. Face validity was assessed by five MHCPs regarding the relevance, ambiguity, and difficulty of the questions, and minor changes were made to the items. Content validity was assessed by calculating the content validity index (CVI) and content validity ratio (CVR). To calculate the CVI, an expert panel (n = 10) in the fields of midwifery, reproductive health, obstetrics and gynecology, health education and promotion, and health service management was asked to rate each item based on the criterion of relevance on a four-point Likert scale. The item-level content validity index (I-CVI) and the scale-level content validity index (S-CVI/Ave) were calculated. According to the recommendation of Polit et al., S-CVI/Ave ≥ 0.9 is considered acceptable [41]. The S-CVI/Ave values for knowledge, attitude, and practice were 0.91, 0.90, and 0.97, respectively. Then, to calculate the CVR, the panel was asked to evaluate the necessity of each item based on the three-point Likert scale "essential", "useful but not essential", and "not necessary". According to the Lawshe table, a minimum CVR value of 0.62 was considered acceptable [42] (S3 Appendix). To determine reliability, the questionnaire was administered to 30 MHCPs twice at intervals of two weeks, and the intra-class correlation coefficient (ICC) and Cronbach's alpha/Kuder-Richardson coefficient were calculated (ICC for knowledge, attitude, and practice were 0.76, 0.82, 0.84, and the Kuder-Richardson for knowledge, and the Cronbach's alphas for attitude and practice were 0.70, 0.73, and 0.71, respectively).

## Scoring

Bloom's cut-off points are recommended to determine the levels of knowledge, attitudes, and practices: good (scores 80–100%), moderate (scores 60–79%), and poor (scores less than 60%) [43]. A three-item scale was used to score the knowledge items. The answer "Yes" was given a score of 1 and the answer "No" or "Do not know" was given a score of zero. The knowledge score for each participant ranged from 0 to 11. Participants' knowledge was classified as good (80–100%; 10–11 points), moderate (60–79%; 8–9 points), and poor (less than 60%; < 8 points). The attitude and practice items were assessed using a five-point Likert scale (strongly agree to strongly disagree and always to never, respectively). Item 5 of attitude and the items 1 to 11 of practice were reverse scored. The expected scores were 13–65 for attitude, and 14–70 for practice. For attitude, the scores were classified as good (80–100%; 52–65 points), moderate (60–79%; 39–51 points), and poor (less than 60%; < 39 points). Similarly, practice was classified as good (80–100%; 56–70 points), moderate (60–79%; 42–55 points), or poor (less than 60%; < 42 points). In addition, the mean score of knowledge, attitudes, and practices was calculated based on 100, which means that the sum of the items for each scale was divided by their maximum score and then multiplied by 100. Moreover, the knowledge, attitude, and practice scores for the mistreatment categories were presented as frequencies with corresponding percentages.

## Statistical analysis

The socio-demographic characteristics of the participants were analyzed using descriptive statistics (including frequencies, percentages, means, and standard deviations). Skewness and kurtosis were used to examine the normal distribution of data [44]. Knowledge and attitude followed normality, and an independent samples t-test and one-way analysis of variance (ANOVA) were used to examine differences in the means of the knowledge and attitude scores by socio-demographic variables. Practice did not follow normality; therefore, the Mann-Whitney U and Kruskal-Wallis tests were used to examine differences in the median of the practice score by socio-demographic variables. Binary logistic regression analysis was used to identify the socio-demographic characteristics associated with knowledge and attitudes. Knowledge and attitude scores were classified as having a binary outcome of good (code: 1; scores of 60% and more) versus poor (code: zero; scores less than 60%) [43]. As the frequency of participants' practice was at a poor level of 8 (3.1%), this outcome was not included in the logistic regression analysis. Variables with a p-value < 0.2 in bivariate regression analysis were selected as candidates for multiple logistic regression. The crude odds ratio (COR) and adjusted odds ratio (AOR) with a 95% confidence interval (CI) were estimated to show the strength of the association. Data were analyzed using SPSS version 22.0 (IBM Corp, Armonk, NY, USA). A p-value of < 0.05 was considered statistically significant.

## Ethical considerations

This study was conducted in accordance with the principles of the Declaration of Helsinki and was approved by the Ethics Committee of Tehran University of Medical Sciences (code number: IR.TUMS.SPH.REC.1400.169). Written informed consent was obtained from all participants, using informed consent form which was approved by the Ethics Committee of Tehran University of Medical Sciences. Permission to mention the names of the hospitals in this study was also obtained from the hospitals.

### Inclusivity in global research

Additional information regarding the ethical, cultural, and scientific considerations specific to inclusivity in global research is included in the S4 Appendix.

## Results

### Socio-demographic characteristics of maternity healthcare providers

A total of 255 participants out of 270 MHCPs working in five public hospitals in Tehran completed the questionnaire (response rate: 94.5%). The mean age of the participants was 31.6 ± 7.7 (ranged, 20–63 years). 138 (54.1%) were married and 100 (39.3%) were residents. Majority of the participants (94.9%) did not have a history of RMC training, and 42.4% of them reported having between 6- to 10-night shifts per month. Other characteristics have been shown in Table 2.

### Maternity healthcare providers' knowledge regarding mistreatment

The mean total knowledge score was 49.4 ± 18.9, out of a total possible score of 100. A total of 211 (82.7%) participants had poor, 39 (15.3%) had moderate, and five (2.0%) had good knowledge regarding mistreatment of women during labour and childbirth (the scoring is explained in the Methods section). Poor knowledge was more apparent in the categories of physical abuse (e.g., applying fundal pressure, where 60.8% of participants did not know that fundal pressure can be experienced as mistreatment), verbal abuse (e.g., threats of poor outcomes and withholding treatment, where 68.6% and 58.8% of participants, respectively, did not know that the threats of poor outcomes and withholding treatment can be experienced as mistreatment), poor rapport between women and providers (e.g., lack of respect for women's preferred birth positions and denial of mobility, where 69.4% and 48.2% of participants, respectively, did not know that these can be experienced as mistreatment), and failure to meet professional standards of care (e.g., frequent vaginal examinations and long delays in service delivery, where 67.5% and 61.2% of participants, respectively, did not know that these can be experienced as mistreatment) (S1 Table).

### Maternity healthcare providers' attitudes regarding mistreatment

The mean total attitude score was 54.6 ± 8.7, out of a total possible score of 100. The attitudes of 177 (69.4%) participants were poor, and 78 (30.6%) were moderate towards mistreatment of women during labour and childbirth (the scoring is explained in the Methods section). Poor attitudes toward items in physical abuse (e.g., applying fundal pressure, where 62.4% of participants believed that fundal pressure could be used to speed up the delivery process if necessary), verbal abuse (e.g., threat, where 53.0% of participants believed that sometimes it is necessary for healthcare providers to force a labouring woman to collaborate with a threat), and discrimination (where 53.8% of participants believed that it is not possible to provide equal services for all women during labour and delivery) had the highest rates (S2 Table).

### Maternity healthcare providers' practices regarding mistreatment

The mean total practice score was 40.4 ± 9.5, out of a total possible score of 100. A total of 247 (96.9%) participants were poor, and eight (3.1%) had moderate levels of practice regarding mistreatment of women during labour and childbirth (the scoring is explained in the Methods section). Verbal abuse (e.g., shouting at woman) was one of the categories in which most participants (n = 147; 57.6%) reported using "always" (1.2%), "often" (8.2%), and "sometimes" (48.2%). In addition, 42.8% of participants reported that they "always" (0.4%), "often" (5.9%), and "sometimes" (36.5%) used fundal pressure (as a form of physical abuse). Moreover,

**Table 2. Socio-demographic characteristics of maternity healthcare providers (n = 255).**

| Characteristics | | n | % |
|---|---|---|---|
| **Age (years)** | | | |
| ≤ 30 | | 136 | 53.3 |
| 31–40 | | 88 | 34.5 |
| > 40 | | 31 | 12.2 |
| Mean ± SD = 31.6 ± 7.7 | | | |
| **Marital status** | | | |
| Single | | 117 | 45.9 |
| Married | | 138 | 54.1 |
| **Profession** | | | |
| Student | Midwifery | 23 | 9.0 |
| | Medical intern | 20 | 7.8 |
| | First year resident | 39 | 15.3 |
| | Second year resident | 24 | 9.4 |
| | Third year resident | 23 | 9.0 |
| | Fourth year resident | 14 | 5.5 |
| Midwife | | 99 | 38.8 |
| Obstetrician | | 13 | 5.1 |
| **Monthly income (IRR)** | | | |
| No income | | 23 | 9.0 |
| 10 000 000–50 000 000 | | 118 | 46.3 |
| 60 000 000–100 000 000 | | 88 | 34.5 |
| 110 000 000–150 000 000 | | 14 | 5.5 |
| > 150 000 000 | | 12 | 4.7 |
| **Work experience (years)** | | | |
| No work experience | | 96 | 37.6 |
| 1–5 | | 81 | 31.8 |
| 6–10 | | 31 | 12.2 |
| > 10 | | 47 | 18.4 |
| Mean ± SD = 5.1 ± 6.8 | | | |
| **Night shifts (per month)** | | | |
| 0 | | 51 | 20.0 |
| 1–5 | | 27 | 10.6 |
| 6–10 | | 108 | 42.4 |
| > 10 | | 69 | 27.1 |
| Mean ± SD = 7.3 ± 4.4 | | | |
| **Trained on RMC** | | | |
| Yes | | 13 | 5.1 |
| No | | 242 | 94.9 |

SD: Standard Deviation; IRR: Iranian Rials; RMC: Respectful Maternity Care

slapping the thighs during delivery as another form of physical abuse is "sometimes" (11.4%) done by them (S3 Table).

## Comparison of socio-demographic characteristics and knowledge, attitude, and practice scores of maternity healthcare providers

Comparison of socio-demographic characteristics and the knowledge, attitude, and practice scores of the MHCPs is shown in Table 3. The knowledge score was significantly ($p < 0.001$)

**Table 3. Comparison of socio-demographic characteristics and knowledge, attitude, and practice scores of the maternity healthcare providers (n = 255).**

| Characteristics | n | Knowledge score | | Attitude score | | Practice score | |
|---|---|---|---|---|---|---|---|
| | | Mean ± SD | p-value | Mean ± SD | p-value | Mean ± SD | p-value |
| **Age*** | | | | | | | |
| ≤ 30 | 136 | 47.1 ± 17.3 | 0.068 | 56.2 ± 8.6 | 0.003 | 41.0 ± 9.5 | 0.039 |
| 31–40 | 88 | 50.9 ± 19.4 | | 53.5 ± 8.5 | | 40.9 ± 9.7 | |
| > 40 | 31 | 55.1 ± 22.8 | | 51.0 ± 8.4 | | 36.1 ± 7.9 | |
| **Marital status**** | | | | | | | |
| Single | 117 | 47.6 ± 19.1 | 0.167 | 55.7 ± 8.9 | 0.070 | 40.7 ± 9.5 | 0.541 |
| Married | 138 | 50.9 ± 18.7 | | 53.7 ± 8.4 | | 40.1 ± 9.5 | |
| **Profession*** | | | | | | | |
| Student | 143 | 46.7 ± 18.2 | < 0.001 | 55.7 ± 8.6 | 0.003 | 43.1 ± 9.9 | < 0.001 |
| Midwife | 99 | 50.7 ± 18.2 | | 54.1 ± 8.5 | | 37.0 ± 7.5 | |
| Obstetrician | 13 | 67.8 ± 21.1 | | 47.4 ± 7.7 | | 36.3 ± 9.3 | |
| **Monthly income (IRR)*** | | | | | | | |
| No income | 23 | 51.3 ± 14.6 | 0.001 | 52.2 ± 7.6 | 0.002 | 38.1 ± 7.9 | < 0.001 |
| 10 000 000–50 000 000 | 118 | 45.6 ± 19.1 | | 56.6 ± 8.3 | | 44.1 ± 9.9 | |
| 60 000 000–100 000 000 | 88 | 52.0 ± 16.2 | | 53.8 ± 8.7 | | 36.8 ± 7.9 | |
| 110 000 000–150 000 000 | 14 | 46.1 ± 26.7 | | 52.0 ± 9.3 | | 36.9 ± 5.7 | |
| > 150 000 000 | 12 | 66.6 ± 21.6 | | 49.1 ± 9.6 | | 37.9 ± 9.3 | |
| **Work experience*** | | | | | | | |
| No work experience | 96 | 47.8 ± 17.8 | 0.037 | 56.4 ± 8.7 | 0.006 | 42.4 ± 9.7 | < 0.001 |
| 1–5 | 81 | 48.0 ± 18.1 | | 55.2 ± 8.5 | | 41.7 ± 9.7 | |
| 6–10 | 31 | 57.4 ± 17.6 | | 51.5 ± 7.4 | | 38.1 ± 8.3 | |
| > 10 | 47 | 51.6 ± 21.9 | | 52.1 ± 8.9 | | 35.5 ± 7.6 | |
| **Night shifts*** | | | | | | | |
| 0 | 51 | 55.6 ± 17.7 | 0.064 | 50.7 ± 7.0 | 0.003 | 35.9 ± 7.3 | 0.002 |
| 1–5 | 27 | 47.8 ± 18.6 | | 55.1 ± 9.9 | | 40.7 ± 9.9 | |
| 6–10 | 108 | 48.5 ± 18.0 | | 55.5 ± 8.6 | | 41.4 ± 9.6 | |
| > 10 | 69 | 46.7 ± 20.5 | | 56.1 ± 8.7 | | 42.0 ± 9.9 | |
| **Trained on RMC**** | | | | | | | |
| Yes | 13 | 61.5 ± 14.4 | 0.018 | 50.2 ± 7.9 | 0.062 | 38.0 ± 8.1 | 0.445 |
| No | 242 | 48.7 ± 18.9 | | 54.9 ± 8.7 | | 40.5 ± 9.6 | |

\* One-way analysis of variance (ANOVA) for knowledge and attitude, and Kruskal-Wallis test for practice;

\*\* Independent samples t-test for knowledge and attitude, and Mann-Whitney U test for practice; p < 0.05; IRR: Iranian Rials; RMC: Respectful Maternity Care

higher in obstetricians (67.8 ± 21.1) than midwives (50.7 ± 18.2) and students (46.7 ± 18.2). MHCPs with 6 to 10 years of work experience had higher knowledge scores (57.4 ± 17.6) than their counterparts (51.6 ± 21.9, 48.0 ± 18.1, and 47.8 ± 17.8, respectively). In addition, the knowledge score was significantly higher among MHCPs who had a monthly income of more than 150 million RLS (66.6 ± 21.6), and MHCPs who had a history of training (61.5 ± 14.4). The attitude score was significantly (p < 0.01) higher in students (55.7 ± 8.6) than in midwives (54.1 ± 8.5) and obstetricians (47.4 ± 7.7). MHCPs who had no work experience (56.4 ± 8.7), and MCHPs with incomes between 10 and 50 million RLS (56.6 ± 8.3) also had a significantly (p < 0.01) higher attitude score. Practice scores were better among students (43.1 ± 9.9), MCHPs who had no work experience (42.4 ± 9.7), and those with incomes between 10 and 50 million RLS (44.1 ± 9.9). More findings are presented in Table 3.

**Table 4. Bivariate and multiple logistic regression analysis for factors associated with maternity healthcare providers' knowledge and attitudes regarding mistreatment (n = 255).**

| Characteristics | Knowledge | | | | Attitude | | | |
|---|---|---|---|---|---|---|---|---|
| | COR (95% CI) | p-value | AOR (95% CI) | p-value | COR (95% CI) | p-value | AOR (95% CI) | p-value |
| **Age** | | | | | | | | |
| ≤ 30 | 0.38 (0.19–0.76) | 0.006 | 0.51 (0.20–1.31) | 0.16 | 2.21 (1.27–3.85) | 0.005 | 1.93 (0.93–4.0) | 0.07 |
| > 30 | 1 | | 1 | | 1 | | 1 | |
| **Marital status** | | | | | | | | |
| Single | 0.62 (0.31–1.21) | 0.16 | 1.01 (046–2.22) | 0.96 | 1.26 (0.74–2.16) | 0.38 | - | - |
| Married | 1 | | 1 | | 1 | | - | |
| **Profession** | | | | | | | | |
| Student | 0.42 (0.21–0.82) | 0.01 | 0.64 (0.22–1.87) | 0.42 | 1.89 (1.08–3.29) | 0.02 | 1.43 (0.65–3.15) | 0.37 |
| Midwife/Obstetrician | 1 | | 1 | | 1 | | 1 | |
| **Monthly income (IRR)** | | | | | | | | |
| ≤ 100 000 000 | 0.34 (0.14–0.82) | 0.01 | 0.63 (0.23–1.72) | 0.37 | 2.62 (0.87–7.89) | 0.08 | 1.20 (0.35–4.14) | 0.76 |
| > 100 000 000 | 1 | | 1 | | 1 | | 1 | |
| **Work experience** | | | | | | | | |
| ≤ 5 | 0.36 (0.18–0.70) | 0.003 | 0.88 (0.24–3.16) | 0.84 | 2.58 (1.34–4.98) | 0.004 | 0.99 (0.32–3.08) | 0.99 |
| > 5 | 1 | | 1 | | 1 | | 1 | |
| **Night shifts** | | | | | | | | |
| ≤ 5 | 1.54 (0.78–3.04) | 0.21 | - | - | 0.43 (0.22–0.81) | 0.01 | 0.45 (0.22–0.89) | 0.02 |
| > 5 | 1 | | - | | 1 | | 1 | |
| **Trained on RMC** | | | | | | | | |
| Yes | 2.24 (0.65–7.64) | 0.19 | 2.35 (0.64–8.59) | 0.19 | 0.39 (0.08–1.83) | 0.23 | - | - |
| No | 1 | | 1 | | 1 | | - | |

1: Reference Category; COR: Crude Odds Ratio; AOR: Adjusted Odds Ratio; CI: Confidence Interval; IRR: Iranian Rials; RMC: Respectful Maternity Care

## Association between socio-demographic characteristics and knowledge and attitudes of maternity healthcare providers

Table 4 shows the factors associated with MHCPs' knowledge and attitudes regarding mistreatment of women during labour and childbirth. According to the bivariate logistic regression analysis, socio-demographic characteristics, including age, marital status, profession, monthly income, work experience, and RMC training history, were significantly associated with knowledge regarding mistreatment, and age, profession, monthly income, work experience, and number of night shifts were factors associated with attitudes regarding mistreatment.

In multiple logistic analysis, MHCPs who had training history on RMC were 2.35 times more likely than those who had no training history to have good knowledge regarding mistreatment, although this was not statistically significant (AOR = 2.35, 95% CI = 0.64–8.59). Single MHCPs were also 1.01 times more likely than their married counterparts to have good knowledge regarding mistreatment; however, this was not statistically significant, as shown in Table 4 (AOR = 1.01, 95% CI = 0.46–2.22). Moreover, the results showed that only the number of night shifts was significantly associated with the attitudes of MHCPs. Thus, MHCPs who had night shifts ≤ 5 per month were 55% less likely to have good attitudes toward mistreatment than those who had night shifts > 5 per month to have good attitudes toward mistreatment (AOR = 0.45, 95% CI = 0.22–0.89). Although other characteristics were not significantly associated with attitudes, it was found that MHCPs with an age ≤ 30 years, students, and a

monthly income $\leq$ 10 million RLS were respectively 1.93, 1.43, and 1.2 times more likely to have good attitudes toward mistreatment than MHCPs with an age $>$ 30 years, midwives/obstetricians, and a monthly income $>$ 10 million RLS (AOR = 1.93, 95% CI = 0.93–4.0; AOR = 1.43, 95% CI = 0.65–3.15; and AOR = 1.20, 95% CI = 0.35–4.14).

## Discussion

This study assessed the KAP of healthcare providers regarding mistreatment of women during labour and childbirth in public teaching hospitals in Tehran, Iran. MHCPs showed critical gaps in knowledge, harmful attitudes, and practices related to physical abuse, verbal abuse, poor rapport between women and providers, failure to meet professional standards of care, and discrimination.

The results of our study showed that most providers did not consider the application of fundal pressure during delivery as a form of mistreatment and believed that it could be used to speed up the delivery process, if necessary. Based on available evidence, fundal pressure during vaginal birth is very common in hospitals [13, 45–49]. A systematic review showed that the prevalence of fundal pressure ranged from 0.6 to 69.2% globally [46]. In Iran, fundal pressure during birth has been reported by women [50]. Although providers described this practice as a way to help women speed up the birth process, it is known as a harmful procedure and an abusive intervention during childbirth that can contribute to a negative experience of childbirth and poor quality of care [45, 47]. However, fundal pressure was commonly self-reported by the participants in our study and was possibly related to poor knowledge about harm and negative attitudes. In addition, participants in our study commonly reported slapping the thighs of a woman during birth, which previous studies also reported physical abuse, such as beatings, slapping, and pinching, by both women [9, 40, 51, 52] and healthcare providers [53–55]. Although some providers believe that it is acceptable to use these tactics to encourage women to collaborate in the birth process and achieve positive health outcomes for mothers and babies, others describe them as immoral and unacceptable [54, 56].

In this study, most providers lacked knowledge of verbal abuse (e.g., threats) as a form of mistreatment. Additionally, most believed that the threat could be used to attract women's collaboration. Regarding practice, most reported that they would shout at women during labour and birth if they did not collaborate. There is extensive evidence of verbal abuse, including shouting, harsh language, insults, intimidation, and threats of non-service or withholding treatment by healthcare providers [18, 26, 35, 54, 57, 58], which aligns with the results of this study. In addition, the results of our study showed that there was poor knowledge about rapport between women and providers, as well as meeting professional standards of care. Most providers did not consider the lack of freedom to choose birth positions, denial of mobility during labour, frequent vaginal examinations, and long delays in providing services as mistreatment. In a study in Iran, not being allowed to choose birth positions and not being allowed to move during labour were the most common forms of mistreatment reported by women [59]. In a study by Shakibazadeh et al., 99.7% of Iranian women reported that they were not allowed mobility during labour and birth companions [22]. The results of a study by Deki et al. also showed that providers' knowledge and practices were weak in some aspects of RMC, including respect for women's preferential birth positions [26]. Therefore, increasing the knowledge of MHCPs for the operationalization of RMC is essential.

The results of our study showed unfavorable attitudes of most providers regarding discrimination, as they believed that it was not possible to provide equal services for all women during labour and birth. Discrimination in access to and experience with services has been shown in

various studies, including discrimination based on race or ethnicity, religion, HIV status, age, and parity [45, 57, 60].

Moridi et al. (2022) showed that Iranian midwives have appropriate knowledge and practice of RMC. In this study, the highest knowledge and practice scores were in the "providing safe care" domain and the lowest knowledge and practice scores were in the "preventing mistreatment" domain [61]. The results of the study by Haghdoost et al. (2021) also showed that Iranian midwives had good knowledge and fair performance towards RMC [62], which is not consistent with our study. This inconsistency may be due to differences in study populations and data collection tools. In these studies, healthcare providers included only midwives [61, 62], whereas our study included all MHCPs (obstetricians, midwives, obstetric residents, midwifery, and medical intern students). Owing to the presence of obstetricians and obstetric residents in Iranian public hospitals, midwives are less involved in vaginal births; they play a more supportive role, and obstetric residents manage most births under the supervision of obstetricians. Therefore, the KAP of our participants may represent a more reliable estimate for healthcare providers caring for women in Tehran.

Our study showed a significant association between the number of night shifts and attitudes towards mistreatment. MHCPs with fewer night shifts were less likely to exhibit positive attitudes towards mistreatment. Hajizadeh et al.'s study showed that night shifts were associated with increased D&A of women [59]. The findings of this study also showed the need to pay attention to MHCPs without a training history on RMC, married, older age groups, midwives/obstetricians, and those with high income.

## Implications for practice and research

Our findings demonstrate clear gaps in the knowledge and attitudes of healthcare providers regarding mistreatment of women during labour and childbirth and can serve as a call to action for maternal and reproductive health policymakers and planners in Iran to improve the quality of maternity care. Continuing education courses should be designed and implemented to increase knowledge, strengthen positive attitudes, and modify practices of MHCPs on mistreatment in Iranian public hospitals. These courses can empower MHCPs to promote and support their rights and those of patients. It is also suggested that RMC should be included in the educational curriculum for obstetric residents and midwifery students.

Future qualitative research should explore MHCPs' and women's perceptions and experiences of mistreatment during labor and childbirth. Moreover, each finding is worth considering when developing related interventions to promote quality and respectful care for women.

## Strengths and limitations

We elicited the KAP of a wide range of obstetricians, midwives, obstetric residents, midwifery, and medical intern students and had a high rate of participation, which may be representative of the total population of MHCPs in Tehran. However, this study had some limitations. First, MHCPs' practices were assessed using a self-reported questionnaire, which may have caused bias owing to the sensitivity of the mistreatment of women during childbirth, and providers did not report their actual practices and/or behaviors. We attempted to reduce this bias by reassuring the participants' anonymity and confidentiality. However, assessing the practice of MHCPs in observational studies is also important. Second, this study was conducted in public teaching hospitals in Tehran; therefore, the results may not be generalizable to all public and private hospitals in Iran.

## Conclusion

The knowledge and attitudes of our participants regarding maternity mistreatment were poor. A small percentage of the participants reported mistreatment practices. Significant gaps were identified in the KAP related to the categories of physical abuse, verbal abuse, poor rapport between women and providers, failure to meet professional standards of care, and discrimination. Among socio-demographic characteristics, the number of night shifts was associated with attitudes toward mistreatment. These findings have important implications for the development of effective interventions to reduce mistreatment of women during labour and childbirth in Iran. These interventions should include designing and implementing continuing education courses and revising educational curricula to increase knowledge, strengthen positive attitudes, and modify the practice of MHCPs.

## Supporting information

**S1 Appendix. Characteristics and names of study hospitals.**
(DOCX)

**S2 Appendix. Questionnaire.**
(DOCX)

**S3 Appendix. Content validity of the questionnaire.**
(DOCX)

**S4 Appendix. PLOS questionnaire on inclusivity in global research.**
(DOCX)

**S1 Table. Knowledge about mistreatment among maternity healthcare providers.**
(DOCX)

**S2 Table. Attitudes about mistreatment among maternity healthcare providers.**
(DOCX)

**S3 Table. Practices about mistreatment among maternity healthcare providers.**
(DOCX)

**S1 Data.**
(XLSX)

## Acknowledgments

This study was part of a PhD dissertation. The authors would like to thank the Taleghani, Mahdieh, Arash, Hazrat Rasoul Akram, and Valiasr hospitals in Tehran, as well as all healthcare providers who participated in this study.

## Author Contributions

**Conceptualization:** Marjan Mirzania, Elham Shakibazadeh, Meghan A. Bohren, Farah Babaey, Sedigheh Hantoushzadeh, Abdoljavad Khajavi, Abbas Rahimi Foroushani.

**Data curation:** Marjan Mirzania.

**Formal analysis:** Marjan Mirzania, Elham Shakibazadeh.

**Funding acquisition:** Marjan Mirzania, Elham Shakibazadeh, Abbas Rahimi Foroushani.

**Investigation:** Marjan Mirzania, Elham Shakibazadeh, Farah Babaey, Sedigheh Hantoushzadeh.

**Methodology:** Marjan Mirzania, Elham Shakibazadeh, Meghan A. Bohren.

**Project administration:** Marjan Mirzania, Elham Shakibazadeh, Meghan A. Bohren.

**Resources:** Marjan Mirzania, Elham Shakibazadeh.

**Software:** Marjan Mirzania, Elham Shakibazadeh, Abbas Rahimi Foroushani.

**Supervision:** Marjan Mirzania, Elham Shakibazadeh, Meghan A. Bohren.

**Validation:** Marjan Mirzania, Elham Shakibazadeh, Meghan A. Bohren, Farah Babaey, Sedigheh Hantoushzadeh, Abdoljavad Khajavi, Abbas Rahimi Foroushani.

**Writing – original draft:** Marjan Mirzania, Elham Shakibazadeh.

**Writing – review & editing:** Marjan Mirzania, Elham Shakibazadeh, Meghan A. Bohren, Farah Babaey, Sedigheh Hantoushzadeh, Abdoljavad Khajavi, Abbas Rahimi Foroushani.

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
