## [Decision Letter · Decision Letter 0]

17 Oct 2023

PONE-D-23-19930Knowledge, attitude and practice of healthcare providers on mistreatment of women during labour and childbirth: a cross-sectional study in Tehran, Iran, 2021PLOS ONE

Dear Dr. Shakibazadeh,

Thank you for submitting your manuscript to PLOS ONE. After careful consideration, we feel that it has merit but does not fully meet PLOS ONE’s publication criteria as it currently stands. Therefore, we invite you to submit a revised version of the manuscript that addresses the points raised during the review process.

We look forward to receiving your revised manuscript.

Kind regards,

Joyce Jebet Cheptum

Academic Editor

PLOS ONE

Journal Requirements:

2. Please include a complete copy of PLOS’ questionnaire on inclusivity in global research in your revised manuscript. Our policy for research in this area aims to improve transparency in the reporting of research performed outside of researchers’ own country or community. The policy applies to researchers who have travelled to a different country to conduct research, research with Indigenous populations or their lands, and research on cultural artefacts. The questionnaire can also be requested at the journal’s discretion for any other submissions, even if these conditions are not met.  

Please find more information on the policy and a link to download a blank copy of the questionnaire here: https://journals.plos.org/plosone/s/best-practices-in-research-reporting. Please upload a completed version of your questionnaire as Supporting Information when you resubmit your manuscript.

4. Please ensure that you include a title page within your main document. You should list all authors and all affiliations as per our author instructions and clearly indicate the corresponding author.

**Additional Editor Comments:**

There is need to indicate in the abstract that this is part of an implementation study. The document should also be well edited

Reviewers' comments:

Reviewer's Responses to Questions

**Comments to the Author**

1. Is the manuscript technically sound, and do the data support the conclusions?

Reviewer #1: Partly

Reviewer #2: Partly

Reviewer #3: Yes

2. Has the statistical analysis been performed appropriately and rigorously? 

Reviewer #1: No

Reviewer #2: N/A

Reviewer #3: Yes

3. Have the authors made all data underlying the findings in their manuscript fully available?

Reviewer #1: Yes

Reviewer #2: Yes

Reviewer #3: Yes

4. Is the manuscript presented in an intelligible fashion and written in standard English?

Reviewer #1: No

Reviewer #2: Yes

Reviewer #3: No

5. Review Comments to the Author

Reviewer #1: this topic was done previously in Iran with a well-developed scale, however the researcher design and use another scale!

The introduction section does not have accurate citations; it is recommended that all the sentence should be cited. Also, sentences and paragraphs have no conceptual connection.

In the method's section, it is determined that they use random stratification, however, in the lines 145-146 showed that they did not select hospitals randomly. Also, the participants were not determined in this section. The MHCPs is not clear that refers to which kinds of providers? Another concern about the accuracy of this study is that how the scale was developed and is it suitable for all kind of health workers? The scale's items are not completely consistent with the content extracted in table 1.

All findings were not discussed appropriately.

Table 7 is not clear and may be confusing for the audiences.

The balance between the number of words in different sections of the manuscript is not observed.

Reviewer #2: This paper was well written with clear aims, but a few strong rationales underlying the importance of the topic. In addition, some problems should be addressed to improve the manuscript:

- While the paper build upon a KAP of providers related to mistreatment of women during childbirth, it is not provided a sufficient and specific information about Exams & Procedures: Informed consent, exposed (vaginal exam,…), confidential information. These items are very important indexes in RMC. Also, it is better to describe an unsupportive birth environment items such pain relief, neglect and also clean up blood ….

- This manuscript must contain Ethics approval and consent to participate, funding and acknowledgement . It is better to give a permission from hospitals to mention the names of hospitals in this article.

Reviewer #3: Title:

The word "Practice" used in the title seem not to match with the outcome. Knowledge and attitude of mistreatment of women during labour and childbirth is in line.

Abstract

1. "Most participants (82.7%) had poor knowledge about physical abuse, verbal abuse, poor rapport between women and providers, and failure to meet professional standards of care". 82.7% is not "most" authours should consider using "majority")

Is the 82.7% poor knowledge relating each of the items listed; verbal abuse, poor rapport between? etc

"Self-reported practices of different types of mistreatment were not common and only 3.1% of the participants were in moderate level". This statement is not clear, authors could consider rephrasing

"These interventions should include designing and implementing continuing education courses and revising the educational curriculums to increase knowledge, strengthen positive attitudes, and modify practice

of maternity healthcare providers, overcoming staff shortages, paying staff fairly, establishing support culture for mother-centered and respectful care, and increasing quality of maternity care". This statement is long to follow. Kindly revise. Authors could consider ending at health care providers; then continue other interventions may include...............

Introduction

Line 79....However, reports of disrespectful maternity care are on 79

the rise worldwide [10-12]. Can this be further explained? why the rise?

"In a study of 38 hospitals and health centers in Ethiopia, 74% of 80

women reported experiencing some forms of mistreatment [13]. A systematic review of five 81

studies in Ethiopia, Kenya, Nigeria and Tanzania found that the prevalence of D&A ranged from 82

15 to 98% [14]. A WHO-led study (2019) in Ghana, Guinea, Nigeria and Myanmar found that 83

35.4% of women experienced physical or verbal abuse, or stigma or discrimination during 84

childbirth [15]". What reasons why attributed to prevalence in these referenced studies?

"A main step in designing interventions to prevent 109

mistreatment is to assess the level of knowledge, attitudes and practices (KAP) of MHCPs, with 110

a goal to improve negative behaviors and reinforce positive behaviors" Is this statement authors idea? If not, kindly reference.

Methods

Sample size: The parameters used in the sample sample size calculation is not clear. eg. mean KAP is questionable. Mean knowledge of course can be determined but combined KAP mean, I don't think so. Authors should provide further explanation and should kindly provide reference of the mean and SD values. Again in the abstract, authors stated total enumeration, how is this related to the sample size calculation. This will need further elaboration.

line 143: "The sampling method was random stratification" Is this for participants or sites?

line 194 "Some items 194 had reverse scoring". Which of the items had these?

Was the score categorization for attitudes also based on Bloom's cut-off point? otherwise this should be referenced.

"Due to abnormality of knowledge and practice scores (p = 0.01 and p = 0.04, 204

respectively), we used Mann-Whitney U and Kruskal-Wallis tests" What was these tests specifically used for, apart from the mere mentioning of abnormality? It serve readers well if detailed analysis is presented, as it stands, it looks scanty

Results

Line 219.....33.8% is not "most". Authors should kindly me mindful of these terms, most...majority etc.same applies here "Most of them had permanent job (n=54; 21.2%) ."

Was knowledge, practice and attitude scores expressed in percentage? Authors should provide readers with detailed explanation.

MHCPs attitudes regarding mistreatment: Results here are presented in freq (%). Authors failed to write about this in the analysis (summary statistics was used to present demographic information, mentioned in the analysis). Same applies to practice...kindly check these and revise.

Table 6 and 7 seem to be presenting the same information. Authors could stick to one. Again, description of table 6 could be summarized with significant predictor variables. No need to present all details as it makes reading boring ...kindly present few and refer readers to the table.

6. PLOS authors have the option to publish the peer review history of their article (what does this mean?). If published, this will include your full peer review and any attached files.

Reviewer #1: No

Reviewer #2: No

Reviewer #3: No

---

## [Author Response · Author response to Decision Letter 0]

12 Nov 2023

Dear reviewers, 

I would like to express our sincere thanks for the detailed and constructive feedback and comments received on our paper. We have benefited a lot from those points and revised our manuscript based on the comments. Below, I would like to share our responses to the comments. We hope that you will find the edited version with satisfaction. 

Reviewer 1

1. This topic was done previously in Iran with a well-developed scale, however the researcher design and use another scale! 

a. The scale developed in Iran includes items about midwives’ knowledge and practice of respectful maternity care (RMC) and not the mistreatment of women during childbirth1. Evidence shows that RMC and mistreatment of women during childbirth are not the same concept, but occupy two extremes of a continuum. The absence or lessening of mistreatment during childbirth does not guarantee RMC for women and newborns during childbirth2, 3. To the best of our knowledge, there is no scale to examine maternity healthcare providers' (MHCPs) knowledge, attitudes, and practices regarding mistreatment during childbirth. Therefore, in this study, a self-report questionnaire prepared using the typology of mistreatment of women during childbirth by Bohren et al. and a literature review was used. References: 1) Moridi M, Pazandeh F, Potrata B. Midwives’ knowledge and practice of Respectful Maternity Care: a survey from Iran. BMC Pregnancy and Childbirth, 2022; 22: 752; 2) Shakibazadeh E, Namadian M, Bohren MA, Vogel JP, Rashidian A, Nogueira Pileggi V, et al. Respectful care during childbirth in health facilities globally: a qualitative evidence synthesis. BJOG, 2018; 125(8):932-42; 3) Afulani PA, Diamond-Smith N, Phillips B, Singal S, Sudinharaset M. Validation of the person-centered maternity care scale in India. Reproductive Health, 2018; 15:147.

2. The introduction section does not have accurate citations; it is recommended that all the sentence should be cited. Also, sentences and paragraphs have no conceptual connection.

a. Thank you for your recommendation. We rephrased the sentences and paragraphs for conceptual connection. We also added references to the sentences. Page 3; Line: 59; Page 4; Lines: 67-74, Lines: 79-82, and Lines: 84-86

3. In the method's section, it is determined that they use random stratification; however, in the lines 145-146 showed that they did not select hospitals randomly. 

a. We rephrased the sampling method. Page 7; Line: 148; Page 8; Lines: 149-152

4. Also, the participants were not determined in this section. The MHCPs is not clear that refers to which kinds of providers?

a. In the “Setting and participants” sub-section, it is noted that MHCPs included obstetricians, midwives, obstetric residents, midwifery, and medical interns. Page 6; Lines: 131-134

5. Another concern about the accuracy of this study is that how the scale was developed and is it suitable for all kind of health workers? 

a. In this study, we first prepared a questionnaire using the typology of mistreatment of women during childbirth presented by Bohren et al. and a literature review. Next, we evaluated the validity and reliability of the scale. Face validity was assessed by five maternity healthcare providers (obstetricians, midwives, obstetric residents, midwifery, and medical interns) regarding the relevance, ambiguity, and difficulty of the questions, and minor changes were made to the items. Content validity was assessed by calculating the content validity index (CVI) and content validity ratio (CVR) by an expert panel (n = 10) in the fields of midwifery, reproductive health, obstetrics, health education and promotion, and health services management. To determine reliability, the questionnaire was administered to 30 maternity healthcare providers twice at intervals of two weeks, and the intraclass correlation coefficient (ICC) and Cronbach's alpha coefficient were calculated. Page 8; Lines: 163-171; Page 9; Lines: 172-194

6. The scale's items are not completely consistent with the content extracted in table 1. 

a. Table 1 provides information about the study hospitals. While the scale's items include categories of mistreatment during childbirth (physical abuse, verbal abuse, stigma and discrimination, failure to meet professional standards of care, poor rapport between women and providers, and health systems conditions and constraints) (Table 2).

7. All findings were not discussed appropriately. 

a. We made some elaborations in the discussion section. Page 19; Lines: 318-320; Page 20; Lines: 356-359

8. Table 7 is not clear and may be confusing for the audiences. 

a. Table 7 (which is currently referred to as Table 4 in the paper) shows the association of sociodemographic characteristics and knowledge, attitude, and practice scores of the participants. We summarized the description of this table, and referred readers to the table. Page 15; Lines: 282-302; Page 16; Lines: 303-307

9. The balance between the numbers of words in different sections of the manuscript is not observed. 

a. We have summarized the results section by moving some tables to the Supporting information section.

Reviewer 2

1. This paper was well written with clear aims, but a few strong rationales underlying the importance of the topic. In addition, some problems should be addressed to improve the manuscript. 

a. We sincerely appreciate your feedback. We rephrased the sentences and paragraphs to mention the importance of the issue in paragraphs 4 and 5. Page 5; Lines: 92-102 and Lines: 103-115

2. While the paper build upon a KAP of providers related to mistreatment of women during childbirth, it is not provided a sufficient and specific information about Exams & Procedures: Informed consent, exposed (vaginal exams, …), confidential information. These items are very important indexes in RMC. Also, it is better to describe an unsupportive birth environment items such pain relief, neglect and also clean up blood …. 

a. In the questionnaire (S1 Appendix), item 7 of knowledge related to “vaginal exams”, item 5 of attitude related to “informed consent”, and item 6 of practice related to “confidentiality of information”. “Pain relief” was also mentioned in item 9 of knowledge and item 8 of practice, and “neglect, abandonment, or long delays” were mentioned in item 3 of knowledge, item 8 of attitude, and item 9 of practice.

3. This manuscript must contain Ethics approval and consent to participate, funding and acknowledgement. It is better to give permission from hospitals to mention the names of hospitals in this article. 

a. Ethical approval and consent to participate, and acknowledgement in the text are mentioned. As required by the journal, we entered the Funding Information section in the Financial Disclosure section of the Submission System. Permission to mention the names of the hospitals was also obtained in this paper, and we added them in the text. Page 11; Lines: 227-229; Page 12; Lines: 230-234; Page 23; Lines: 426 and Page 24; Lines: 427-429

Reviewer 3

1. The word “Practice” used in the title seems not to match with the outcome. Knowledge and attitude of mistreatment of women during labour and childbirth is in line. 

a. In our study, practices scale was assessed using 14 items, and the results showed that only 3.1% of the participants reported moderate mistreatment practices towards birthing women. However, like many other cross-sectional studies, we assessed the practices in our study using self-report scale, which may have caused bias owing to the sensitivity of the mistreatment of women during childbirth. We attempted to reduce this bias by reassuring the participants’ anonymity and confidentiality. However, assessing the practices of MHCPs in an observational study will also be important. We added this to the study limitations section. Page 22; Lines: 401-406

2. “Most participants (82.7%) had poor knowledge about physical abuse, verbal abuse, poor rapport between women and providers, and failure to meet professional standards of care”. 82.7% is not “most” authours should consider using “majority”). Is the 82.7% poor knowledge relating each of the items listed; verbal abuse, poor rapport between? etc 

a. We rephrased the sentence for better comprehension. Page 2; Lines: 33-38

3. “Self-reported practices of different types of mistreatment were not common and only 3.1% of the participants were in moderate level”. This statement is not clear, authors could consider rephrasing 

a. We rephrased the sentence for better comprehension. Page 2; Lines: 38-40

4. “These interventions should include designing and implementing continuing education courses and revising the educational curriculums to increase knowledge, strengthen positive attitudes, modify the practice of maternity healthcare providers, overcoming staff shortages, paying staff fairly, establish a support culture for mother-centered and respectful care, and increase the quality of maternity care”. This statement is long to follow. Kindly revise. Authors could consider ending at health care providers; then continue other interventions may include...... 

a. Thank you for pointing this out. We revised the statement accordingly. Page 3; Line: 48 and Line: 50

5. Line 79.... However, reports of disrespectful maternity care are on the rise worldwide [10-12]. Can this be further explained? Why the rise? 

a. We rephrased this sentence in the text. Page 4; Lines: 84-86

6. “In a study of 38 hospitals and health centers in Ethiopia, 74% of women reported experiencing some forms of mistreatment [13]. A systematic review of five studies in Ethiopia, Kenya, Nigeria and Tanzania found that the prevalence of D&A ranged from 15 to 98% [14]. A WHO-led study (2019) in Ghana, Guinea, Nigeria and Myanmar found that 35.4% of women experienced physical or verbal abuse, or stigma or discrimination during childbirth [15]”. What reasons why attributed to prevalence in these referenced studies? 

a. We edited some sentences in the Introduction section. Reference to the prevalence in these studies is provided to further explain the sentence (.... growing evidence shows that mistreatment during childbirth has become a common experience for women worldwide [12-15]). Page 4; Lines: 84-91

7. “A main step in designing interventions to prevent mistreatment is to assess the level of knowledge, attitudes and practices (KAP) of MHCPs, with a goal to improve negative behaviors and reinforce positive behaviors”. Is this statement author's idea? If not, kindly reference. 

a. Yes; this statement is the author's idea.

8. The parameters used in the sample size calculation are not clear. e.g. mean KAP is questionable. Mean knowledge of course can be determined but combined KAP mean, I don't think so. Authors should provide further explanation and should kindly provide reference of the mean and SD values. Again in the abstract, author's stated total enumeration, how is this related to the sample size calculation. This will need further elaboration. 

a. We provided further explanations for the parameters used in the sample size calculation, and added references to the mean and SD values. We also edited the results section in the abstract. Page 7; Lines: 143-147; Page 2; Line: 33

9. Line 143: “The sampling method was random stratification” Is this for participants or sites?

a. We rephrased the sampling method for better comprehension. Page 7; Line: 148; Page 8; Lines: 149-152

10. Line 194 “Some items had reverse scoring”. Which of the items had these? 

a. We added it in the text. Page 11; Lines: 206-207

11. Was the score categorization for attitudes also based on Bloom's cut-off point? Otherwise this should be referenced. 

a. Yes; the score categorization for knowledge, attitudes, and practices was based on Bloom's cut-off point. We rephrased the sentence for better comprehension. Page 10; Lines: 199-201

12. “Due to abnormality of knowledge and practice scores (p = 0.01 and p = 0.04, respectively), we used Mann-Whitney U and Kruskal-Wallis tests”. What was these tests specifically used for, apart from the mere mentioning of abnormality? It serve readers well if detailed analysis is presented, as it stands, it looks scanty 

a. We revised the statistical analysis. Page 11; Lines: 219-224

13. Line 219..... 33.8% is not “most”. Authors should kindly me mindful of these terms, most...majority etc. same applies here “Most of them had permanent job (n=54; 21.2%).” 

a. Thank you for pointing this out. We revised the sentences accordingly. Page 12; Lines: 242-244

14. Was knowledge, practice and attitude scores expressed in percentage? Authors should provide readers with detailed explanation. 

a. We added explanations about this in the Methods section (Scoring and Statistical analysis). Page 11; Lines: 211-215

15. MHCPs attitudes regarding mistreatment: Results here are presented in freq (%). Authors failed to write about this in the analysis (summary statistics was used to present demographic information, mentioned in the analysis). Same applies to practice...kindly check these and revise. 

a. We revised the statistical analysis. Page 11; Lines: 217-225

16. Table 6 and 7 seem to be presenting the same information. Authors could stick to one. Again, description of table 7 could be summarized with significant predictor variables. No need to present all details as it makes reading boring ... kindly present few and refer readers to the table. 

a. Table 6 shows the practices of providers regarding mistreatment of women during labour and childbirth, and Table 7 (which is currently listed as Table 4 within the paper) shows the association of sociodemographic characteristics and knowledge, attitude, and practice scores of providers. We summarized the description of this table, and refer readers to the table. Page 15; Lines: 282-302; Page 16; Lines: 303-307

---

## [Decision Letter · Decision Letter 1]

11 Jun 2024

PONE-D-23-19930R1Knowledge, attitude and practice of healthcare providers on mistreatment of women during labour and childbirth: a cross-sectional study in Tehran, Iran, 2021PLOS ONE

Dear Dr. Shakibazadeh,

Thank you for submitting your manuscript to PLOS ONE. After careful consideration, we feel that it has merit but does not fully meet PLOS ONE’s publication criteria as it currently stands. Therefore, we invite you to submit a revised version of the manuscript that addresses the points raised during the review process. Your manuscript has been assessed by one of the three original reviewers - please see the reviewer's comments below. The reviewer still has some outstanding concerns, including the sample size determination, and requests further clarification on several aspects of the manuscript, including statistical tests for normality. Could you please revise the manuscript to carefully address the concerns raised?

We look forward to receiving your revised manuscript.

Kind regards,

Steve Zimmerman, PhD

Senior Editor, PLOS ONE

Journal Requirements:

Reviewers' comments:

Reviewer's Responses to Questions

**Comments to the Author**

1. If the authors have adequately addressed your comments raised in a previous round of review and you feel that this manuscript is now acceptable for publication, you may indicate that here to bypass the “Comments to the Author” section, enter your conflict of interest statement in the “Confidential to Editor” section, and submit your "Accept" recommendation.

Reviewer #3: (No Response)

2. Is the manuscript technically sound, and do the data support the conclusions?

Reviewer #3: Yes

3. Has the statistical analysis been performed appropriately and rigorously? 

Reviewer #3: Yes

4. Have the authors made all data underlying the findings in their manuscript fully available?

Reviewer #3: Yes

5. Is the manuscript presented in an intelligible fashion and written in standard English?

Reviewer #3: Yes

6. Review Comments to the Author

Reviewer #3: Abstract

Line 34: Most participants (82.7%) had poor knowledge about physical abuse, verbal abuse, poor rapport between women 35 and providers, and failure to meet professional standards of care. The use of “most” seems inappropriate, I will suggest “majority” instead of “most”. Does the 82.7% poor knowledge refer to each of the domains “ physical abuse, verbal abuse, poor rapport between women and providers, and failure to meet professional standards of care”. This is not clear.

Line 28: All maternity healthcare providers (obstetricians, midwives) and students were invited to the study… “All eligible maternity healthcare providers (obstetricians, midwives) and students were invited to the study”.

Line 36: Most participants (69.4%) were 36 alright with physical abuse, verbal abuse, and discrimination. This statement is not also clear. What does authors mean by “alright”.

Conclusion: The conclusion seem too long, authors should revise and go straight to conclude study findings.

Introduction

There are no information on “knowledge and practices of mistreatment of healthcare providers”. A Brief information on the this main outcome will be important in the background.

Methods

Line 139-142: “Considering that the mean 139 scores of KAP of participants were calculated on the basis of 100, the standard deviation of the 140 score was 16.7 and the number of samples was determined with 95% confidence level (Z1- α 2 = 141 1.96) and a maximum error of two scores estimated this mean” how did authors come by the values used in the calculation of sample size?. If this is a previous study, provide reference.

“Due to normal attitude 205 scores (p = 0.17), we used independent samples t-test and one-way (ANOVA)” what exactly was this used for?. Authors should expand on this.

Which people were involved in the data collection, how were they trained?

Line 212- 213, “This study did not require the respondents to identify themselves and for confidentiality, ID was used instead of names for the questionnaires.” This information may not be necessary.

Results

Line 219: 33.8% is not most, this may need revision.

The use of “most”, “majority” need to be used meticulously in the entire work. I suggest authors consider when appropriate to use these descriptive words.

Line 244: the scoring is explained “in” the Methods section

7. PLOS authors have the option to publish the peer review history of their article (what does this mean?). If published, this will include your full peer review and any attached files.

Reviewer #3: No

---

## [Author Response · Author response to Decision Letter 1]

15 Jun 2024

Dear Editor, 

I would like to express our sincere thanks for the detailed and constructive feedback and comments received on our paper. We have benefited a lot from those points and revised our manuscript based on the comments. Below, I would like to share our responses to the comments in the table. We hope that you will find the edited version with satisfaction. 

The reviewer’s comments 

1. In the abstract, line 34: Most participants (82.7%) had poor knowledge about physical abuse, verbal abuse, poor rapport between women 35 and providers, and failure to meet professional standards of care. The use of “most” seems inappropriate, I will suggest “majority” instead of “most”. Does the 82.7% poor knowledge refer to each of the domains “physical abuse, verbal abuse, poor rapport between women and providers, and failure to meet professional standards of care”. This is not clear.

Response: We rephrased the sentence for better comprehension. Page 2; Lines: 33-36

2. In the abstract, line 28: All maternity healthcare providers (obstetricians, midwives) and students were invited to the study… “All eligible maternity healthcare providers (obstetricians, midwives) and students were invited to the study”. 

Response: Thank you for pointing this out. We revised the statement accordingly. Page 2; Line: 26-28

3. In the abstract, line 36: Most participants (69.4%) were 36 alright with physical abuse, verbal abuse, and discrimination. This statement is not also clear. What does authors mean by “alright”.

Response: We rephrased the sentence for better comprehension. Page 2; Lines: 37-38

4. In the abstract, conclusion: The conclusion seem too long, authors should revise and go straight to conclude study findings.

Response: We summarized the conclusion. Page 3; Lines: 44-48

5. In the introduction, there are no information on “knowledge and practices of mistreatment of healthcare providers”. A Brief information on the this main outcome will be important in the background. 

Response: We provided the information in page 5, lines 99-108. 

6. In the methods, line 139-142: “Considering that the mean scores of KAP of participants were calculated on the basis of 100, the standard deviation of the score was 16.7 and the number of samples was determined with 95% confidence level (Z1- α 2 = 1.96) and a maximum error of two scores estimated this mean” how did authors come by the values used in the calculation of sample size?. If this is a previous study, provide reference. 

Response: We provided further explanations for the parameters used in the sample size calculation, and added references to the mean and SD values. Page 7; Lines: 139-144; Page 11; Lines: 209-211.

7. “Due to normal attitude scores (p = 0.17), we used independent samples t-test and one-way (ANOVA)” what exactly was this used for? Authors should expand on this. 

Response: We revised the statistical analysis. Page 11; Lines: 217-222

8. Which people were involved in the data collection, how were they trained?

Response: We added explanations about this in the Data collection section. Page 8; Lines: 151-152. Page 8; Lines: 153-156; 

9. Line 212- 213. “This study did not require the respondents to identify themselves and for confidentiality, ID was used instead of names for the questionnaires.” This information may not be necessary. 

Response: We removed the sentence. Page 12; Line: 299

10. In the results, line 219: 33.8% is not most, this may need revision. 

Response: Thank you for pointing this out. We revised the sentence accordingly. Page 12; Lines: 238-240.

11. The use of “most”, “majority” need to be used meticulously in the entire work. I suggest authors consider when appropriate to use these descriptive words. 

Response: We revised the use of these terms in the entire text. Page 2; Lines: 33-36; Page 2; Lines: 37-38; Page 12; Lines: 238-240

12. Line 244: the scoring is explained “in” the Methods section 

Response: We added it in the text. Page 13; Line: 248.

---

## [Decision Letter · Decision Letter 2]

30 Jul 2024

PONE-D-23-19930R2Knowledge, attitude and practice of healthcare providers on mistreatment of women during labour and childbirth: a cross-sectional study in Tehran, Iran, 2021PLOS ONE

Dear Dr. Shakibazadeh,

Thank you for submitting your manuscript to PLOS ONE. After careful consideration, we feel that it has merit but does not fully meet PLOS ONE’s publication criteria as it currently stands. Therefore, we invite you to submit a revised version of the manuscript that addresses the points raised during the review process.

We look forward to receiving your revised manuscript.

Kind regards,

Fereshteh Behmanesh, PhD

Academic Editor

PLOS ONE

Journal Requirements:

Reviewers' comments:

Reviewer's Responses to Questions

**Comments to the Author**

1. If the authors have adequately addressed your comments raised in a previous round of review and you feel that this manuscript is now acceptable for publication, you may indicate that here to bypass the “Comments to the Author” section, enter your conflict of interest statement in the “Confidential to Editor” section, and submit your "Accept" recommendation.

Reviewer #3: All comments have been addressed

Reviewer #4: All comments have been addressed

2. Is the manuscript technically sound, and do the data support the conclusions?

Reviewer #3: Yes

Reviewer #4: Partly

3. Has the statistical analysis been performed appropriately and rigorously? 

Reviewer #3: Yes

Reviewer #4: No

4. Have the authors made all data underlying the findings in their manuscript fully available?

Reviewer #3: Yes

Reviewer #4: No

5. Is the manuscript presented in an intelligible fashion and written in standard English?

Reviewer #3: Yes

Reviewer #4: Yes

6. Review Comments to the Author

Reviewer #3: (No Response)

Reviewer #4: Dear Researcher

This manuscript id “Knowledge, attitude and practice of healthcare providers on mistreatment of women during labour and childbirth: a cross-sectional study in Tehran, Iran, 2021

As the authors state, Mistreatment of women during childbirth had a key role in women’s birth experience. Studies in this field are valuable.

- In the data collection section, from lines 151 to 156 written in one sentence, write in several sentences

- It is better to use univariate and multivariate regression in the analysis

7. PLOS authors have the option to publish the peer review history of their article (what does this mean?). If published, this will include your full peer review and any attached files.

Reviewer #3: No

Reviewer #4: No

---

## [Author Response · Author response to Decision Letter 2]

31 Aug 2024

Dear Editor, 

I would like to express our sincere thanks for the detailed and constructive feedback and comments received on our paper. Below, I would like to share our responses to the comments in the table. We hope that you will find the edited version with satisfaction. 

Best regards, 

Dr. Elham Shakibazadeh, 

Corresponding author

The reviewer’s comments :

1. In the data collection section, from lines 151 to 156 written in one sentence, write in several sentences

Response: Thank you for pointing this out. We revised the statement accordingly.

Page 7; Lines: 144-149

2. It is better to use univariate and multivariate regression in the analysis

Response: We included univariate and multiple regression in the analysis.

Page 11; Lines: 213-220

Page 18; Lines: 294-315

Page 19; Lines: 316-318

---

## [Editor Report · Decision Letter 3]

8 Sep 2024

Knowledge, attitude and practice of healthcare providers on mistreatment of women during labour and childbirth: a cross-sectional study in Tehran, Iran, 2021

PONE-D-23-19930R3

Dear Dr. Shakibzadeh,

We’re pleased to inform you that your manuscript has been judged scientifically suitable for publication and will be formally accepted for publication once it meets all outstanding technical requirements.

Kind regards,

Fereshteh Behmanesh, PhD

Academic Editor

PLOS ONE
---

## [Editor Report · Acceptance letter]

24 Sep 2024

PONE-D-23-19930R3 

PLOS ONE

Dear Dr. Shakibazadeh, 

I'm pleased to inform you that your manuscript has been deemed suitable for publication in PLOS ONE. Congratulations! Your manuscript is now being handed over to our production team.

Kind regards, 

on behalf of

Dr. Fereshteh Behmanesh 

Academic Editor

PLOS ONE